# Mobile Banking: An Innovative Solution for Increasing Financial Inclusion in Sub-Saharan African Countries: Evidence from Nigeria

**Alfonso Siano [1,*], Lukman Raimi [2], Maria Palazzo [1] and Mirela Clementina Panait [3]**

[1] Department of Political and Communication Sciences, University of Salerno, 84084 Salerno, Italy; mpalazzo@unisa.it

[2] Department of Entrepreneurship and Management, American University of Nigeria, Lagos 23401, Nigeria; lukman.raimi@aun.edu.ng

[3] Department of Cybernetics, Economic Informatics, Finance and Accounting, Petroleum-Gas University of Ploiesti, 100680 Ploiesti, Romania; mirela.matei@upg-ploiesti.ro

* Correspondence: sianoalf@unisa.it

**Abstract:** Purpose—This research discusses emerging trends in financial inclusion, barriers and factors influencing mobile banking as an innovative solution for increasing financial inclusion in sub-Saharan Africa (SSA) with a specific focus on Nigeria. Design/methodology/approach—Using a qualitative meta-synthesis (QMS), an interpretivist research paradigm, authors provide an analytical tool for understanding the subject of inquiry by integrating findings from previous studies and relevant data from the reports of the Central Bank of Nigeria on emerging trends in financial inclusion. Findings—Three major factors emerged as drivers of mobile banking in Nigeria: (a) the ease of using mobile devices for personal banking transactions including prompt information about users' financial transactions (savings and withdrawals) immediately through SMS (short message service) alert (easy management of my account); (b) the security/safety concerns of theft and cyber fraud; (c) social influence of friends, relatives, policy makers and social trends. Implications—In contextualizing mobile banking in SSA and in Nigeria in particular, this paper contributes to exploring the growth in the use of mobile banking by linking it with the "value in use" (VIU) perspective. This approach of the service dominant logic involves three sub-constructs (experience, personalization, and relationship), which all validate and support the proposed assertion that mobile banking is adopted by users because of utility expectancy (perceived usefulness), effort expectancy (perceived ease of use), and social influence expectancy (opinions of friends/relatives). Originality/value—This research, although qualitative in nature, validates information technology (IT) adoption theories/perspectives and enriches the "value in use" approach.

**Keywords:** financial inclusion; innovative solution; mobile banking; Nigeria; sub-Saharan Africa (SSA); qualitative meta-synthesis (QMS); banking industry; digital transformation; value in use approach

---

## 1. Introduction

The global economy faces extreme poverty, slower growth, climate change, widening inequalities, unemployment, and growing inconstant working conditions, but the plight of sub-Saharan Africa (SSA) is worse because more than 204 million people are unemployed, and the worsening unemployment situation provides breeding grounds for forced labour, slavery and human trafficking [1]. Additionally, the report of the World Bank from 2019 [2] identified extreme poverty, growing public debt/debt risk, slow growth of the labour market, rising labour force, and gender disparities as the critical inhibiting factors holding back economic growth and sustainable development in SSA.

In the midst of the economic crisis explicated above, the people and businesses in the area, including Nigeria, suffer an extremely low level of financial inclusion [3,4]. Studies have explicated that the phenomenon of the lack of access (financial exclusion) to basic financial services is a global problem, but the problem is most pronounced in emerging economies including SSA [5,6].

At the global level, financial inclusion has occupied the attention of international organisations especially the agendas of the millennium development goals (MDGs) and sustainable development goals (SDGs). Some of the targets of the expired MDGs alert the United Nations member countries to the pressing issue of financial inclusion and its complexity. Specifically, the goals of fighting extreme poverty foundationally (MDG1); achieving gender equality to promote equal opportunities for women to access employment, social protection, and training (MDG3); and forging a global partnership for development (MDG 8) directly address financial inclusion challenges. Similarly, the targets of the ongoing SDGs that directly impact on financial inclusion include: no poverty (SDG1), zero hunger (SDG2), good health and wellbeing (SDG3), quality education (SDG4), decent work and economic growth (SDG8), industry, innovation, and infrastructure (SDG9), reduced inequality (SDG10), peace, justice and strong institutions (SDG16) and partnerships for the goals (SDG17).

In Nigeria, several attempts have been made by the government to improve financial inclusion through a number of public-sector led credit schemes and poverty alleviation programmes such as National Economy Reconstruction Fund (NERFUND); the People's Bank, Community Banking Models, the Bank of Industry (BOI), the Microfinance Institutions (MFIs) the Small and Medium Enterprises Equity Investment Scheme (SMEEIS), National Poverty Eradication Programme, Youth Enterprise with Innovation in Nigeria (You Win) Programme, Subsidy Reinvestment and Empowerment Programme or SURE-P, National Enterprise Development Programme or NEDEP and several others [7,8]. In spite of the financial inclusion intervention schemes, the National Financial Inclusion Strategy (drafted in 2012 and revised in 2018) and the "cash-less Nigeria policy", the efforts failed because of the government's inability to properly nurture its development programmes, weak reward system, dysfunctional structures and endemic poor programme implementation [8–11]. In the early 2000s, mobile banking emerged as an information technology (IT)-driven innovative technology, which greatly improved the degree of financial inclusion in the continent [5,12–15]. Mobile banking provides virtual access for individuals and businesses to procure financial transactions such as savings, funds' transfer, and stock market deals with banks at any convenient time and place [16]. The receptiveness to mobile banking is impressive, as the majority of the banks in Nigeria have adopted and introduced mobile banking applications. Additionally, mobile banking has thrived because Nigeria has the fastest growing telecommunication infrastructure in Africa and the third in the world [17]. The benefit of IT infrastructure is supported by the population of over 150 million people [18]. Providing an enabling environment for better and improved financial inclusion, Nigeria justifies the adoption and introduction of mobile banking to strengthen local and international efforts towards financial inclusion (the timely introduction of mobile phone technology in the continent within the last 10 years resulted in 82% mobile banking penetration in Nigeria, the highest penetration of mobile banking across developing markets [19].

It is possible to state that research on financial inclusion in SSA is just emerging [20–22]. To enrich the body of knowledge in this important field of financial inclusion, there is a need to explore empirical evidence. Financial inclusion, in fact, has been delayed in the continent prior to the liberalization of the financial sector in the 1980s, because many African banks were owned, controlled and heavily regulated by the governments as monopolies, a situation that restricted the adoption of innovative technologies in banking operations [3,6,23].

Thus, the purpose of this exploratory qualitative research is to critically discuss the emerging trends in financial inclusion, the barriers to financial inclusion and factors influencing mobile banking adoption as an innovative solution for increasing financial inclusion in SSA with a specific focus on Nigeria.

There are several parts to this paper. The first part presents the introduction of the thematic issue of mobile banking and financial inclusion. The second part focuses on the definition of financial inclusion and mobile banking from both managerial and academic perspectives, examining barriers to financial inclusion and factors influencing adoption of mobile banking in Nigeria. Then, this paper discusses methodology and proposes findings and discussions of thematic issues. The concluding part discusses the implications and recommendations on the mobile banking agenda and direction in sub-Saharan Africa.

## 2. Theoretical Background

### 2.1. Financial Inclusion: Definitions and Features

The need to achieve the sustainable development goals brings to the attention of the authorities a pressing issue, namely the complexity of the phenomenon, the multitude of tools that could be used to achieve the set targets, the large number of stakeholders involved or that may be involved and the lack of financial funds [24–27]. For these reasons, financial inclusion has become important, both at the microeconomic and macroeconomic levels, as it can be a tool for promoting the principles of sustainable development [28–36]. Financial inclusion is not only a concern of financial institutions that are trying to attract more and more categories of consumers and have a responsible attitude towards them but also of public authorities [33,37–42]. In fact, there are specialists who consider financial inclusion, for some countries, even a catalyst for sustainable development, a tool for poverty reduction and a facilitator of economic growth [27,43–45]. Financial inclusion is a complex concept in a continuous evolution taking into account the transformations that are taking place worldwide from an economic, social and political point of view. Financial inclusion refers to "access to and usage of appropriate, affordable, and accessible financial services" [46]. Given the complexity of the phenomenon and the importance of national and regional characteristics on it, financial inclusion presents multiple definitions in the literature, which emphasize certain aspects such as financial innovation, microfinance and the use of mobile phones [47]. According to the World Bank, which is a main international financial institution with preoccupations in this field, financial inclusion is defined as "access to useful and affordable financial products and services that meet their needs—transactions, payments, savings, credit and insurance—delivered in a responsible and sustainable way" [43].

Apiors and Suzuki (2018) consider that financial inclusion supposes "the full range of services (payments, savings, credit, and insurance), to specific quality features of delivery (for example, stability and affordability), inclusiveness (with special focus on the poor), and choice (offer of service by a range of institutions)" [48].

The concept of social inclusion emerged as a solution to the phenomenon of social exclusion that has been identified by geographers who have observed limited access of certain categories of citizens to basic financial services [49]. After the emergence of the concept of social exclusion, the 90s were characterized by intense scientific concerns regarding citizens' access to financial services for payment, savings, and credit, but also to insurance services. Subsequently, the European Commission focused on this issue, considering that financial exclusion is a component of the phenomenon of social exclusion that affects, in various forms, important categories of citizens of the European Union [45]. Financial exclusion targets people that "encounter difficulties accessing and/or using financial services and products in the mainstream market that are appropriate to their needs and enable them to lead a normal social life in the society in which they belong." It must be highlighted that the phenomenon of financial exclusion affects both developed and developing countries, but the share of the non-banked population is different across the world, depending on the level of development of that country [45].

Although the concept of financial exclusion has emerged in the UK [50], the phenomenon mainly affects developing countries characterized by a precarious financial infrastructure from an institutional, technical, legislative, or social point of view. Given the manifestation of the phenomenon of financial exclusion predominantly in developing countries, more and more definitions and approaches [51,52]

focus on: (i) citizens' access to the formal or semi-formal financial sector composed of commercial banks, development finance institutions, post offices, microfinance banks, credit unions and cooperatives, (ii) improving the process of saving but also accessing loans, (iii) enhancing risk management, (iv) developing innovative financial solutions, (v) protection of consumer rights.

The concerns of international financial institutions such as the World Bank, development agencies, national market supervisors and regulators have intensified [53] in order to attract an increasing number of citizens to gain access to products and services. As the financial inclusion is a multidimensional concept [47], various categories of stakeholders from supply and demand sides must be involved. Public authorities and financial institutions must offer sustainable alternatives to consumers, and citizens must make efforts to increase the degree of financial education by participating in programs conducted in this regard by various entities [45,54–58]. Given the multiple crises and scandals that have particularly affected the banking market, financial institutions need to substantially improve their behaviour so as to inspire consumer confidence. Therefore, consumers are also required to use the proper financial services according to their ability to understand financial phenomena or income level [59,60]. In addition, in certain particular situations, such as immigrants, financial consumers have to overcome certain language, cultural, and religious barriers [45,48,61]. Consumers must not only show responsibility when making financial decisions but must also have the ability to learn and adapt to the new conditions of the financial market generated by the digitization of operations. Achieving the objectives set at national level, usually by launching financial inclusion strategies, therefore requires sustained efforts both by the population and by the financial authorities and institutions.

Given the large number of definitions [47], three dimensions of financial inclusion have been established, namely access (refers to physical proximity and affordability), usage (refers to regularity, frequency, duration of time used) and quality (refers to products well-tailored to clients' needs and to appropriate segmentation to develop products for all income levels). To the three dimensions, other specialists [44,62] added an additional dimension—choice (Figure 1).

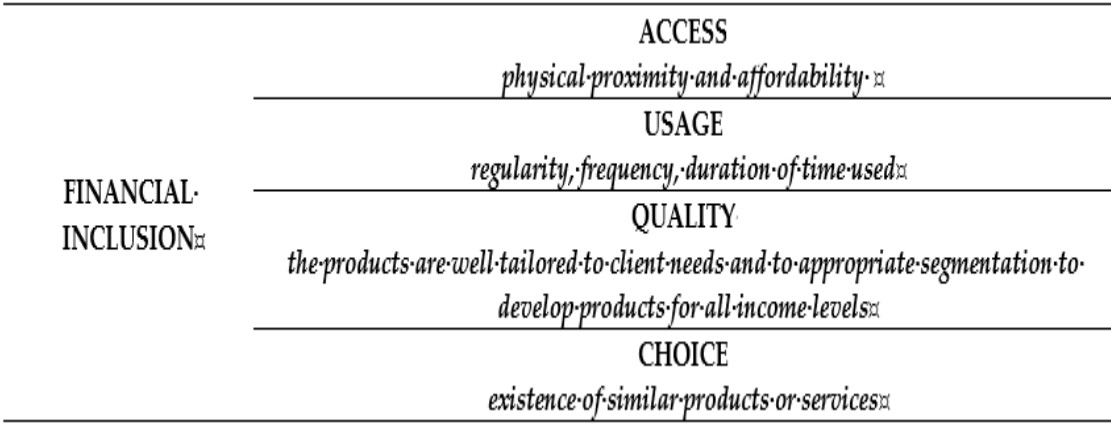

**Figure 1.** Dimensions of Financial Inclusion. Source [47].

Mobile banking largely stimulates and promotes financial inclusion in developing economies, especially in sub-Saharan Africa [13], mobile penetration promotes financial inclusion, and significantly reduces the probability of a household becoming poor [63–69]. Having briefly synthesized this complex topic, this paper will focus on the analysis of financial inclusion, taking into consideration Nigeria. Nigeria is one of the leaders on the African continent along with Kenya in the process of increasing the degree of financial inclusion through the use of mobile banking. In addition, the successful experiences registered in these countries have been models for other countries in the areas of implementation and development of mobile banking systems. Despite the similarities between the two countries in terms of the level of development, the degree of financial exclusion penetration rates in mobile phone usage, the structure of the banking system and the approach to the need to increase financial inclusion were

undertaken with different instruments. The reluctance of the Central Bank of Nigeria to use this model based on a mobile phone company monopoly, the structure of the mobile phone market and the late launch of initiative that coincided with the onset of the 2008 international financial crisis made the results modest. Therefore, the model of "bank-led" (or a non-MNO-led-mobile network operators-led) ecosystem used in Nigeria generates different network externalities compared to mobile network operator systems used in Kenya [70–73].

*2.2. Barriers to Financial Inclusion*

In developing countries, financial exclusion is fuelled by the country's economic structure (mainly agriculture), the location of the majority of the population is in the rural areas with poor banking intermediation and low spread of money deposit bank branches across different parts of the country due to the policy of commercial financial institutions [27]. Other factors that limit the financial inclusion of the population are bureaucracy, high costs of banking products and services and distance [52,74,75].

Leveraging on past studies developed in Nigeria, Table 1 below clearly shows the degree of financial inclusion in the country measured in terms of five metrics, namely (a) % of adult using formal payment systems, (b) % of adult with savings accounts, (c) % of adult with credit from the bank, (d) % of adult with insurance policies, (e) % of adult with pension schemes and (f) % of adult financial exclusion.

**Table 1.** Access to financial services in Nigeria (2010–2018).

| Focus Area | 2010 | 2012 | 2014 | 2016 | 2018 |
|---|---|---|---|---|---|
| % of Adult using Formal Payments System | 22 | 20 | 24 | 38 | 40 |
| % of Adult with Savings Accounts | 24 | 25 | 32 | 36 | 24 |
| % of Adult enjoying Credits | 2 | 2 | 3 | 3 | 2 |
| % of Adult with Insurance Policies | 1 | 3 | 1 | 2 | 2 |
| % of Adult with Pension Schemes | 5 | 2 | 5 | 7 | 8 |
| % of Adult Financial Exclusion | 46.3 | 39.7 | 39.5 | 41.6 | 36.8 |

Source: [76].

The barrier of distance mentioned above is generated both by the weak representation in the territory of commercial banks, but also by the state of infrastructure that exacerbates the problem of moving citizens from rural areas to cities to make various financial operations [77]. Moreover, Table 2 below shows that weak representation of deposit money bank branches engenders the problem of financial inclusion in Nigeria. In fact, states in Northern Nigeria such as Adamawa, Jigawa, Bornu, Bauchi, and Kwara with a large population have very few bank branches, whereas smaller cosmopolitan states with a relatively low population have large bank branches.

**Table 2.** Number of deposit money bank branches in Nigeria by state 2010–2019.

| | 2010 | 2011 | 2012 | 2013 | 2014 | 2015 | 2016 | 2017 | 2018 | 2019 |
|---|---|---|---|---|---|---|---|---|---|---|
| Number of Banks | 24 | 24 | 21 | 24 | 24 | 25 | 25 | 26 | 27 | 29 |
| Branches Abroad | 2 | 2 | 2 | 1 | 1 | 2 | 2 | 2 | 2 | 2 |
| Abia | 146 | 125 | 138 | 147 | 144 | 135 | 142 | 137 | 135 | 135 |
| Abuja (FCT) | 398 | 359 | 379 | 397 | 380 | 369 | 421 | 437 | 382 | 390 |
| Adamawa | 67 | 79 | 63 | 61 | 47 | 47 | 57 | 64 | 66 | 60 |
| Akwa-Ibom | 99 | 92 | 100 | 94 | 92 | 103 | 106 | 114 | 88 | 102 |
| Anambra | 237 | 222 | 228 | 224 | 219 | 218 | 219 | 214 | 209 | 227 |
| Bauchi | 53 | 50 | 46 | 46 | 47 | 48 | 50 | 47 | 55 | 47 |
| Bayelsa | 37 | 37 | 37 | 38 | 38 | 38 | 38 | 39 | 35 | 35 |
| Benue | 75 | 57 | 73 | 76 | 67 | 63 | 69 | 71 | 78 | 65 |
| Borno | 79 | 68 | 71 | 69 | 83 | 72 | 60 | 61 | 58 | 56 |

**Table 2.** *Cont.*

|  | 2010 | 2011 | 2012 | 2013 | 2014 | 2015 | 2016 | 2017 | 2018 | 2019 |
|---|---|---|---|---|---|---|---|---|---|---|
| Cross-River | 79 | 76 | 76 | 80 | 79 | 74 | 78 | 79 | 72 | 75 |
| Delta | 198 | 177 | 194 | 198 | 178 | 180 | 200 | 205 | 183 | 183 |
| Ebonyi | 35 | 45 | 33 | 33 | 59 | 61 | 37 | 36 | 59 | 42 |
| Edo | 183 | 162 | 188 | 192 | 144 | 165 | 178 | 188 | 159 | 177 |
| Ekiti | 80 | 60 | 64 | 76 | 91 | 87 | 86 | 92 | 76 | 77 |
| Enugu | 141 | 116 | 142 | 147 | 158 | 151 | 159 | 162 | 127 | 148 |
| Gombe | 40 | 36 | 36 | 37 | 43 | 41 | 36 | 37 | 65 | 34 |
| Imo | 104 | 97 | 100 | 102 | 110 | 105 | 98 | 100 | 99 | 94 |
| Jigawa | 39 | 37 | 36 | 38 | 63 | 66 | 38 | 36 | 43 | 34 |
| Kaduna | 183 | 170 | 169 | 171 | 154 | 164 | 168 | 173 | 169 | 157 |
| Kano | 193 | 186 | 183 | 183 | 174 | 170 | 178 | 179 | 195 | 161 |
| Katsina | 62 | 55 | 58 | 59 | 73 | 78 | 56 | 55 | 52 | 47 |
| Kebbi | 40 | 40 | 37 | 38 | 95 | 37 | 37 | 35 | 49 | 61 |
| Kogi | 80 | 77 | 82 | 84 | 88 | 80 | 79 | 82 | 70 | 71 |
| Kwara | 79 | 139 | 75 | 79 | 104 | 101 | 78 | 84 | 100 | 85 |
| Lagos | 1766 | 1453 | 1692 | 1678 | 1443 | 1486 | 1645 | 1686 | 1478 | 1624 |
| Nasarawa | 58 | 51 | 49 | 48 | 68 | 69 | 49 | 49 | 67 | 52 |
| Niger | 80 | 76 | 79 | 82 | 67 | 65 | 78 | 86 | 64 | 71 |
| Ogun | 175 | 402 | 161 | 154 | 137 | 142 | 154 | 172 | 153 | 169 |
| Ondo | 121 | 109 | 110 | 119 | 106 | 101 | 113 | 120 | 120 | 112 |
| Osun | 105 | 118 | 101 | 104 | 101 | 99 | 106 | 108 | 86 | 99 |
| Oyo | 236 | 203 | 223 | 237 | 347 | 343 | 222 | 237 | 195 | 223 |
| Plateau | 79 | 72 | 77 | 75 | 75 | 71 | 70 | 67 | 75 | 83 |
| Rivers | 302 | 246 | 310 | 311 | 292 | 275 | 312 | 319 | 275 | 301 |
| Sokoto | 53 | 53 | 52 | 52 | 43 | 45 | 53 | 52 | 60 | 47 |
| Taraba | 37 | 41 | 35 | 35 | 40 | 40 | 34 | 27 | 39 | 30 |
| Yobe | 35 | 35 | 33 | 35 | 38 | 41 | 34 | 31 | 27 | 29 |
| Zamfara | 35 | 33 | 34 | 40 | 39 | 38 | 30 | 31 | 38 | 34 |
| TOTAL | 5809 | 5454 | 5564 | 5639 | 5526 | 5470 | 5570 | 5714 | 5301 | 5437 |

Source: [78].

In addition, the rapid growth of the population in developing countries contributes to the deepening of the phenomenon of population impoverishment, and implicitly of social exclusion. Despite the efforts made by the public authorities in the banking sector and by the credit institutions, the population growth precedes the growth of the banked population, there are also multiple regional differences [45].

The ways of manifesting the phenomenon of financial inclusion and financial literacy differ depending on consumers' level of income and the development level of the countries [44,77,79]. In high-income countries, financial exclusion affects a small part of the population, and citizens have concerns about making sophisticated investments in the financial market and an investment plan for retirement [80]. In low-income countries, the concerns of financial authorities and institutions regarding financial inclusion are much more complex and lasting because they concern a considerable proportion of the population being attracted from the non-formal to the formal sector, and the causes of financial exclusion are generated by the economic situation, and not only the poor financial education of the population [80]. In these countries, citizens have problems when simply trying to open a bank account or purchase insurance.

In developing countries, the main cause of low financial inclusion is extreme poverty that affects a large part of the population. Citizens of these countries do not have bank accounts because they do not have enough money to save [47,52]. Studies conducted for African countries have revealed a positive link between per capita income and financial inclusion. Thus, the study conducted by Alenoghenain (2017) for 15 African countries for the period 2005–2014 showed that a high income per capita generates a higher financial inclusion [81].

Therefore, a phenomenon with multiple economic and social implications such as financial exclusion/inclusion has a complex of determining factors, and solving the problem requires a systemic approach and the involvement of several categories of stakeholders.

In addition, financial exclusion affects not only poor people and women but also migrants [44,82–84]. The trend of international migration has also generated financial problems, as migrants have to overcome certain psychological barriers generated by distrust in complex banking systems (compared to those in the country of origin), lack of financial education specific to developing countries or the time horizon (if the stay is prolonged, there is a change in attitude towards the banking system). The phenomenon of saving specific to migrants is followed by the periodic sending of large sums of money to the family in the country of origin, which is why having a bank account to make bank transfers is essential [85,86]. The frequency of remittances is different, being influenced by the needs of the family, the sex of the migrant, the prospects of opening a business in the country of origin or buying a home and by the intention to reunite the family in the destination country [61,87].

### 2.3. Drivers of Mobile Banking

The adoption of mobile banking has improved all aspects of financial transactions for both corporate and non-corporate customers of banks in Nigeria [21,88,89]. Conceptually, the term mobile banking is understood as the use of mobile devices for undertaking virtual financial transactions (especially savings, funds' transfer, and stock market transactions) with banks by customers at any convenient time and place [16]. Financial consumers are increasingly interested in achieving easy access to banking services and products, either to save money or to pay for goods or to send money to relatives or friends. In this sense, the statistical data provided by Global Findex [90] show the increase in the percentage of the population that makes domestic remittances. The intensification of the international migration process has also led to an increase in the frequency and value of the remittances sent by migrants to origin countries, mobile banking being a feasible solution that allows financial transfers but it also contributes to increasing the degree of financial education and financial inclusion in these countries. The feasibility of this method is also supported by the low costs generated by the intensification of competition between banks. From the consumers' perspective, the future of mobile banking depends on essentials such as ubiquity, instant connectivity, pro-active functionality, convenience, access to the service regardless of time and place, privacy, and savings in time and effort [91–93] (Singh 2012, Tiwari et al., 2016, Akturan and Tezcan, 2012). In addition, "The younger generations of the society seem to be fascinated by modern data and telecommunication services" [92]. The analysis of financial inclusion vis-à-vis mobile banking and information technology (IT) has to be embedded in IT adoption theories/perspectives. According to Bankole et al. (2011), the widely used theories in mobile banking [16] include the technology acceptance model (TAM), the extended technology acceptance model (TAM2), the theory of reasoned action (TRA), the theory of planned behavior (TPB) and the unified theory of use and acceptance of technology (UTAUT). Other specialists have focused on the task–technology fit (TTF) theory and the diffusion of innovation (DOI) in order to explain the determinants of the acceptance of mobile banking [94,95] (Munoz-Leiva et al. 2017, Sharma 2019). The technology acceptance model (TAM), the main influential model, focuses on two main issues—perceived usefulness and perceived ease of use—to explain the variance in users' intentions [96] (Luarn and Pin, 2005). According to Nasri and Charfeddine 2012, the "TPB suggests that in addition to attitudinal and normative influence, a third antecedent to the theory called perceived behavioral control, perceived behavioral (PBC), also influences behavioral intentions and actual behavior" [97]. Particularly, TAM and TPB were used by several researchers to investigate the factors influencing users' behavioural intentions towards mobile banking [16,98]. Summing up the insights from all the IT adoption theories, proponents argued that mobile banking is an off-shoot of IT that is adopted by users for financial transactions because of key factors, namely performance expectancy (perceived usefulness), effort expectancy (perceived ease of use), social influence (opinions of friends,

relatives), facilitating conditions, trust and privacy, convenience and cost, user satisfaction and national culture [98–100].

Starting from these premises, the receptiveness of Nigerian customers towards mobile banking is impressive, as all banks have adopted and introduced mobile banking applications (the popular mobile banking apps in Nigeria include Diamond Mobile Banking, Stanbic IBTC Mobile Banking, First Bank Mobile Banking, Eco Bank Mobile Banking, Access Bank Mobile Banking, Fidelity Bank Mobile Banking and Sterling Bank Mobile Banking). Several factors have been identified by scholars as the determinants influencing mobile banking in the country. In a sequential order, the following are insights on factors influencing the adoption of mobile banking.

Anyasi and Otubu (2009) noted that mobile banking was adopted in Nigeria as a preferred means of accessing financial services with or without access to traditional banks, it also offers a way to lower the costs of transferring funds from place to place, while at the same time improving financial inclusion by bringing unbanked people into contact with the formal financial systems [101]. A decade ago, a result of the research of Oni et al. (2010) indicated that the e-banking system in Nigeria is widely adopted and preferred by customers because it is convenient, easy to use, time efficient, and appropriate for financial transactions. However, the users have concerns for the network security especially the privacy of transactions [102].

Moreover, Bankole et al. (2011) explained that utility expectancy and effort expectancy are major factors influencing behavioural intention to adopt mobile banking in Nigeria. Specifically, the ease of using a mobile device for personal banking transactions (easy management of accounts); the security and safety concerns of moving cash around regarding cyber fraud; and prompt information about users' financial transactions (savings and withdrawals) immediately through the SMS alert. However, the social influence of friends, relatives, colleagues at work and the service providers (banks) is not a major influence on mobile banking users in Nigeria [16].

Nevertheless, Aliyu et al. (2012) contended that the six critical success factors that influence the adoption of mobile banking in Nigeria include awareness, ease of use, security, cost, reduced reluctance to change and accessibility of mobile devices for undertaking financial transactions [103].

Furthermore, Balogun et al. (2013) affirmed that three factors largely influenced customers' satisfaction with different aspects of e-banking in Nigeria, such as telephone banking, mobile banking, point of sale terminals, smart cards, and television banking. These factors are the quality services provided by banks through SMS alerts and e-mail alerts; the second factor is access to the option of electronically opening a bank account, and the third factor is availability of automated teller machines in strategic locations for making withdrawals when needed without going into the banks [104].

However, Adewoyein (2013) reported that the adoption of mobile banking by customers is premised on a number of factors such as transactional convenience, better turn-around time (saving time), quick transaction alert, reduction in service cost and overall customer satisfaction [105]. Similarly, Njoku and Odumeru (2013) opined that seven factors positively influence the adoption of mobile banking in Nigeria, these include relative advantage that mobile banking gives to bank customers; less complexity of mobile technology; compatibility of mobile banking with customers' norms, belief, existing values, and past experience; perceived trialability/ease of mobile banking when experimented with by the customers; perceived observability/visibility of the outcomes of mobile banking to customers; age of customers using mobile banking; and educational qualification of customers using mobile banking [7].

Moreover, Agwu and Carter (2014) noted that the use of mobile phones for mobile banking has fundamentally changed the way Nigerians (as individuals and businesses) conduct financial transactions [106]. Particularly, mobile banking is more established than internet banking and ATM (automated teller machine) services because it has wider reach than both internet banking and mobile banking (effort expectancy).

Three factors identified by Tarhini et al. (2015) as key drivers of mobile banking adoption [107] in Nigeria include: (a) functionality factor that consists of awareness, ease of use and accessibility; (b) risks

factors that comprise trust, security and privacy; and (c) context factor that includes convenience. From the viewpoints of Agu et al. (2016), mobile banking became a prominent feature in banking operations in the country because it is an effective technology for providing the growing population of customers with fast, accessible, reliable and quality services. Secondly, as an innovative option, mobile banking enables users carry out banking transactions anywhere and anytime and it provides an easy platform for paying for goods or services [108].

In the view of Khan and Ejike (2017), widespread adoption of mobile banking is influenced by access to technological domains, which enhance good knowledge regarding mobile devices, but the degree of convenience and satisfaction of usage are very low [109]. Additionally, Bagudu et al. (2017) explained that the singular and most important success factor for the widespread adoption of mobile banking in the country is access to functional mobile technology, which makes it easy for customers to carry out financial transactions on their mobile handsets with ease [110].

In SSA, there has been an increase in both registration and usage of financial inclusion tools such as mobile banking and related systems. This fact is corroborated by Bille et al., (2018) who reported that as of 2018 there has been an encouraging improvement in financial inclusion because unbanked people and the small and medium-sized enterprises (have been well integrated into the financial landscape in sub-Saharan Africa as a result of improved access to financial inclusion tools, which have been made accessible by the financial service providers, especially banks, microfinance institutions, different mobile network operators and licensed payment service providers [111]. Particularly, the World Bank report indicated that financial inclusion has reached over seven million new financial services users across SSA, from Senegal to Tanzania, from Nigeria to Zambia [111] (Bille et al., 2018).

## 3. Methodology

In view of the exploratory nature of this research, a qualitative meta-synthesis (QMS), an interpretivist research paradigm, provides a rich analytical tool for this study.

The qualitative meta-synthesis was applied as a suitable approach and essential stage towards enlarging the research and setting the basis for analysing the contributions of this paper [112]. This step makes it easy to identify boundaries of the conceptual content of the field and contribute to theory advances [113]. In line with the qualitative research tradition, a qualitative meta-synthesis is an analytical approach that is widely used in meta study to integrate different findings from diverse studies on the same subject of inquiry from different contexts [8,114].

The purpose of integrating findings and insights from groups of studies in a qualitative meta-synthesis is to develop an explanatory theory or model that explains the phenomenon being investigated better and richer [115]. With regard to steps involved in carrying out a sound QMS, Walsh and Downe (2005) proposed seven steps for using qualitative meta-synthesis in research [116].

(1) Framing a meta-synthesis exercise: our topic from the outset is framed for a qualitative meta-synthesis exercise.
(2) Locating relevant papers: we searched and located several papers on mobile banking within and outside SSA to gain richer insights on the subject of inquiry.
(3) Deciding what to include: after a literature audit of searched and located papers on mobile banking, we selected those related to mobile banking issues and financial inclusion in SSA in line with the qualitative meta-synthesis tradition.
(4) Appraising studies: the selected papers from SSA were then appraised to draw rich and meaningful information for making informed and evidence-based findings in line with the qualitative meta-synthesis tradition.
(5) Comparing and contrasting exercise: the findings in the selected papers from SSA were compared and contrasted.
(6) Reciprocating translation: we offered an explanation for the similar and opposing findings.

(7)   Synthesizing translation: the mixed findings extracted from different papers selected were then fused and synthesised to give a unique explanation for the trends and direction of mobile banking in relation to financial inclusion in SSA.

These steps were followed in the above study.

First of all, in order to manage the groundwork for the following QMS, and to identify emerging trends in financial inclusion, the key items were defined. In line with the mentioned definitions and features, but extending to related terms, "financial inclusion", "in-debtedness", "microfinance", "digital financial services", "mobile banking", and "Nigerian banks" were used for identifying related peer reviewed journal articles. The main reason for searching with these alternative items/terms was to ensure the comprehensive nature and content validity of our key terms.

The exploration for meaningful publications was conducted through databases such as Wiley, Elsevier, Scopus, Emerald, and Springer. The selection of these databases was based upon their use by academics in past systematic QMS in the fields of business, management, marketing, communication, and social sciences, along with the openness of the data for the analysis. In addition, the analysis was conducted by selecting only papers with a managerial/marketing perspective in peer-reviewed scientific journals published in the English language. Several papers in different languages or with a dissimilar focus were excluded. The list with articles used for this analysis is presented in Appendix A. The list of relevant papers was attained after removing duplicated articles and then exploring the filtered articles. Appendix A also refers to some relevant topics that are analysed in each paper, such as number of factors affecting mobile banking and type of specific factors affecting mobile banking.

Additionally, Appendices B and C show selected articles classified under several categories: number of research publications per year and classification based on number of determinants/factors able to influence mobile banking development. Moreover, for a meaningful selection of scholarly papers, a search for relevant articles on financial inclusion and mobile banking was carried out using a purposive sampling technique. Furthermore, 58 sampled articles that focused on Nigeria were systematically reviewed and synthesized with insights from the reports of the Central Bank of Nigeria to form integrated findings that explain the emerging trends in financial inclusion [117] and the barriers and factors influencing mobile banking in Nigeria.

The analysis and synthesis of papers aimed to reach two main objectives, first of all, briefly summarise existing research by identifying hot themes/matters; afterwards, contribute to shape the conceptual field of study. Obviously, the authors found it challenging to read everything about financial inclusion and mobile banking, especially if it is considered that the topic is not always called this label or the research is not always related to the selected country: Nigeria.

## 4. Findings and Discussions

Leveraging on the qualitative meta-synthesis, the findings that emerged could be classified into emerging trends on financial inclusion and the barriers and factors that influenced the adoption of mobile banking in Nigeria.

On the emerging trends of financial inclusion in connection with mobile banking, the research found that several attempts have been made by the government to improve financial inclusion in the country through a number of public-sector led credit schemes such National Economy Reconstruction Fund (NERFUND); the People's Bank, Community Banking Models, the Microfinance Institutions (MFIs), the Bank of Industry (BOI), the Small and Medium Enterprises Equity Investment Scheme (SMEEIS), National Poverty Eradication Programme, Youth Enterprise with Innovation in Nigeria (You Win) Programme, Subsidy Reinvestment and Empowerment Programme or SURE-P, National Enterprise Development Programme or NEDEP and several others (Table 3).

**Table 3.** Emerging trends in financial inclusion and mobile banking.

| SN | Policies on Financial Inclusion | Governance Level | Target Audience |
|----|----------|----------|----------|
| 1 | National Economy Reconstruction Fund (NERFUND) | National | Individuals and Businesses across Nigeria |
| 2 | People's Bank of Nigeria | National | Individuals, petty traders, artisans and small businesses across Nigeria |
| 3 | Community Banking Models | National | Individuals, petty traders, artisans and small businesses |
| 4 | Microfinance Institutions (MFIs) | National | Individuals, petty traders, artisans and small businesses |
| 5 | Bank of Industry (BOI) | National | Corporate entities—SMEs (small and medium-sized enterprises) across Nigeria |
| 6 | Small and Medium Enterprises Equity Investment Scheme (SMEEIS) | National | Corporate entities—SMEs across Nigeria |
| 7 | National Poverty Eradication Programme (NAPEP) | National | Individuals, petty traders, artisans and small businesses |
| 8 | Youth Enterprise with Innovation in Nigeria (You Win) Programme | National | Individuals, petty traders, artisans and small businesses |
| 9 | Subsidy Reinvestment & Empowerment Programme (SURE-P) | National | Individuals, petty traders, artisans and small businesses |
| 10 | Millennium Development Goals (MDGs) | International | National institutions, People and businesses |
| 11 | Sustainable Development Goals (SDGs) | International | National institutions, People and businesses |

Source: authors, based on summary of reviewed literature on barriers to financial inclusion.

At the global level, financial inclusion has also occupied the attention of international organisations especially the agendas of the millennium development goals (MDGs) and sustainable development goals (SDGs). Some of the targets of the expired MDGs and SDGs alert the United Nations member countries to the pressing issue of financial inclusion and its complexity. Providing an enabling environment for better and improved financial inclusion in Nigeria justifies the introduction of mobile banking.

Considering the barriers to financial inclusion, this research identified a number of institutional and environmental barriers (Table 4). These barriers provided enabling grounds and springboards for the introduction of mobile banking in Nigeria. Financial inclusion intervention schemes failed because of the government's inability to properly nurture its development programmes, weak reward system, dysfunctional structures, and endemic poor programme implementation. Related to the barriers above are issues of bureaucracy of financial operations, high costs of banking products and services and distance of banks to the population.

With regard to drivers of mobile banking, this research identified three major factors as influencers of mobile banking in Nigeria; the ease of using a mobile device for personal banking transactions including prompt information about users' financial transactions (savings and withdrawals) immediately through the SMS alert (easy management of account); the security and safety concerns of moving cash around regarding cyber fraud; social influence of friends, relatives, policy makers and social trends. With regard to the contextualization of the findings, the three factors that emerged could be categorized as (a) utility expectancy (perceived usefulness), (b) effort expectancy (perceived ease of use), and (c) social influence expectancy (opinions of friends, relatives). Each of the three factors has specific elements assigned to them as shown in Table 5.

**Table 4.** Barriers to financial inclusion (summary of reviewed literature on barriers to financial inclusion).

| SN | Nature of Barrier | Barrier Classification |
|---|---|---|
| 1 | Government's inability to properly nurture its financial inclusion interventions and programmes | Institutional factor |
| 2 | Dysfunctional structures and endemic poor programme implementation | Institutional factor |
| 3 | Structure of the economy (Agriculture-based economy) | Institutional factor |
| 4 | Location of the majority of the population | Environmental factor |
| 5 | Bureaucracy of financial operations | Environmental factor |
| 6 | High costs of banking products and services | Environmental factor |
| 7 | Distance of banks to the population. | Environmental factor |
| 8 | Number of money deposit bank branches | Environmental factor |

Source: authors, based on summary of reviewed literature on barriers to financial inclusion.

**Table 5.** Contextualizing the factors influencing mobile banking (summary of reviewed literature on drivers of mobile banking).

| SN | Value in Use (VIU)Sub-Constructs | Determinants/Drivers of Mobile Banking | Main Specific Factor Elements |
|---|---|---|---|
| 1 | Experience | Utility expectancy | Prompt, transaction notification, Trust and privacy, Satisfaction using mobile banking |
| 2 | Personalization | Effort expectancy | Convenience and cost, Ease of management, personal banking transactions |
| 3 | Relationship | Social influence expectancy | Influence of advert, opinions of friends and relatives, behavioural influence of people on mobile banking, institutional policy on cashless policy, other pressures |

Source: authors, based on summary of reviewed literature on barriers to financial inclusion.

These three key factors seem to be in line with the complex perspective of the "value co-created in use", a very well-known facet of the service dominant logic (SDL) [118]. This approach considers value as co-created in use (VIU) because consumers assess and decide the value of a proposition based on their usage [119]. The VIU involves three sub-constructs: experience, personalization, and relationship.

Taking into consideration the first item, experience indicates an impressive, cognitive and/or emotional interface that creates essential value [118]. In our case, utility expectancy (perceived usefulness) can be perceived as a factor that is able to generate value for customers pushing them to use the banking sector's services as they are strongly based on setting a positive experience for these kind of potential clients through mobile banking.

On the other hand, personalization highlights the distinctiveness of the usage process, the value developed by individual needs [118]. In the current analysis, this is expressed by the effort expectancy (perceived ease of use). In fact, due to the fact that mobile banking is seen by customers as feasible and not too complicated, this also affects the perception that consumers have of banking services in general, before they were considered as misleading and problematic; nowadays, this trend is changing, setting the place for a new horizon in the sector.

Finally, relationship considers a mutual, continuing exchange and alliance not only among consumers, but also between the company and its clients, following an active communication setting [118]. The sub-construct of relationship, expressed in our study by the factor of social influence expectancy (opinions of friends, relatives), is able to empower current and potential consumers to resolve daily problems; thus, engendering mobile banking with a value that was not taken into account by people that until now decided to not use any kind of financial product [120].

## 5. Implications, Limitations and Future Research

This paper has the merit to deepen the understanding of current trends in financial inclusion. This topic can be considered nowadays as a catchphrase for banking specialists, researchers and other categories of stakeholders considering its implications on economic growth and achieving the SDGs [51,88,121]. Financial inclusion can be considered as an international challenge [122] but on the African continent, this issue is more stringent as there are high rates of financial exclusion and in addition, the heterogeneity generated by the specific economic situation, ethnic warfare, religious considerations and cultural perceptions is very high [123]. Little is known about emerging trends in financial inclusion and mobile banking, barriers to financial inclusion and factors influencing mobile banking especially in SSA and particularly in Nigeria. Exploring the trends, factors, barriers, and main items that impact on financial inclusion is essential, particularly in the African context where the level of financial inclusion is extremely low. Thus, this paper has contributed to the existing literature as it focuses on the importance of mobile banking as an innovative solution for increasing financial inclusion in a specific sub-Saharan African country: Nigeria.

Connecting back to previous conceptual research on VIU and SDL, a key theoretical contribution of this study lies in its extension of the boundary of this approach. Our study empirically illustrates the effects of VIU and of its variables in the field of mobile banking. In line with Ranjan and Read (2016) [118], our results empirically validate that consumers' VIU is due to personalized, memorable experiences and positive relationships with mobile banking features and applications. The three factors presented, in fact, not only increase consumers' propensity to continue to use mobile banking for the valuable services it provides to them but also nurtures a solid loyalty towards the banking sector that was never so developed before in the selected area.

From a managerial point of view, our findings regarding the applicability of SDL and the importance of VIU in mobile banking can be of special interest to managers and mobile app designers. Our results highlight that VIU is essential to boost financial inclusion towards banking services and financial products. From the VIU perspective, in fact, consumers are dynamically engaged in the value co-creation process, which relies on experience, personalization and relationship. This statement sets a challenge for managers and mobile app designers because for a client to play a role in the co-creation process, she/he have not only to download and utilize the app but log into the app for practitioners to attain her/his suggestions/comments/opinions. For instance, managers have to create more chances to co-create VIU with the client because the bank/financial institution needs to propose appropriate customized services and customer care that produce better experiences and set the basis for long term relationships with the customers. Additionally, leveraging on personalization is another key asset for managers, banks have to "push" clients to try new personalised financial services through mobile banking apps, offering to pioneer several incentives in order to reward their positive attitude towards the organisation and its services.

Moreover, this paper states that financial inclusion is a complex phenomenon and the solutions to reduce financial exclusion are economic, technical and social and have to be used in a mixed set up by public authorities and financial institutions.

For this reason, our work shows policy makers that the strategy of increasing financial inclusion must be based on the collaboration between different categories of stakeholders such as financial consumers, credit institutions, public authorities with supervisory and control attributions in financial markets, schools and universities.

From this perspective, this paper suggests that policy makers play an important role in creating online communities for consumers that use mobile banking. They could foster financial inclusion practices in virtual places, guiding users and acting as a reference point, or simply answer questions and solve doubts that consumers may have regarding using the financial products. In fact, policy makers should act as a "filter" between banks and users. This involves them being the first to catch customers' specific requests and needs and presenting them to banking managers.

Additionally, they can be useful in explaining how to use more complex software to banking clients who are now choosing to employ only services with a low level of complexity; in this case, policy makers can show to users that a major degree of personalization in services is not going to harm them, instead it should be preferred as it can give them more benefits. Actually, if they are involved in presenting these new personalised services, users could be more interested in approaching them as they trust these kind of players more than employees or spokespersons who are directly paid by banks and other financial institutions.

Having said that, these individuals should be rewarded for their important contribution inside the community with tax incentives, financial benefits, easy-terms loan, soft financing, etc. Actually, feedback posted in online communities can be also appreciated by managers, who can follow the users' point of view to find new ways to increase the level of personalization of services offered through mobile banking.

In addition, public service announcements and other kinds of social advertising should be created by policy makers to communicate to the public that financial inclusion (spread thanks to mobile banking) can highly benefit not only the economy of a country but also the whole society's lifestyle and level of education.

On the other hand, it must be highlighted that our analysis shows several limitations that provide some precious opportunities to future researchers in this underestimated area of research.

We acknowledge that the highlighted features of financial inclusion, explored in this paper, do not represent a comprehensive list of factors expressing all the potentialities hidden inside this concept as we took into consideration a specific country and its peculiarities. This proposes opportunities for further analyses in this vast area employing other specific methodologies and context-related factors.

Each of the labels chosen to conduct the qualitative meta-synthesis are strictly linked with the concept of financial inclusion but, of course, they are not part of an exhaustive list. Therefore, an increase in the labels used may help in the future to better understand how the concept has evolved through the years. Given the different approaches of central banks in Nigeria and Kenya to mobile money services, one of the areas that future studies should investigate is the adequacy of existing financial regulations and policies in SSA regarding mobile banking, particularly because mobile banking is an interface between financial services and telecoms. Another important raging issue that should be explored in the future is the moderating effect of culture on the relationship between mobile banking and financial inclusion in developing countries. Finally, as the financial inclusion represents a challenging area of research that is continuously evolving, we were unable to include all features of this topic in our analysis. In fact, what is clear now is that the main characteristics of financial inclusion may vary over time as they are highly affected by other factors and by technological developments. Therefore, we encourage future researchers to conduct a longitudinal analysis by including a wider range of factors, especially those related to the online world. Considering the intensification of the financial innovation process and the increase in the standard of living, consumers are oriented towards more and more complex products and services. For this reason, a future direction of research may be towards mobile financial services as a tool for financial inclusion. Such efforts will surely further contribute to the theoretical development of this research area.

## 6. Towards Conclusions

This study discusses the emerging trends in financial inclusion and the barriers and factors influencing mobile banking as an innovative solution for increasing financial inclusion in SSA with a specific focus on Nigeria. After a qualitative meta-synthesis of the literature and other secondary materials, it was found that mobile banking was introduced by the government and adopted by major banks in the country to strengthen national and international efforts towards financial inclusion in spite of institutional and environmental barriers engendering financial inclusion. Three major factors emerged from the qualitative meta-synthesis as drivers of mobile banking in Nigeria. Firstly, the ease of using a mobile device for personal banking transactions including prompt information

about users' financial transactions (savings and withdrawals) immediately through the SMS alert (easy management of account). Secondly, the security and safety concerns of theft and of cyber fraud. Thirdly, social influence of friends, relatives, policy makers and social trends.

In contextualizing mobile banking in sub-Saharan Africa in general and Nigeria in particular, it could be stated that the growth of the use of mobile banking is largely supported by the IT adoption theories/perspectives, such as the technology acceptance model (TAM) and the extended technology acceptance model (TAM2), which all validate and support the assertion that technology is adopted by users because of (a) utility expectancy (perceived usefulness), (b) effort expectancy (perceived ease of use), and (c) social influence expectancy (opinions of friends, relatives).

**Author Contributions:** Conceptualization, A.S., L.R., M.P., M.C.P., methodology, L.R.; formal analysis, A.S., L.R., M.P., M.C.P., resources, A.S., L.R., M.P., M.C.P.; data curation, L.R.; writing—original draft preparation, A.S., L.R., M.P., M.C.P.; writing—review and editing, A.S., L.R., M.P., M.C.P.; visualization, A.S., L.R., M.P., M.C.P.; supervision, A.S., L.R., M.P., M.C.P.; project administration, A.S. All authors have read and agreed to the published version of the manuscript.

**Funding:** This research received no external funding.

**Acknowledgments:** Although the views and ideas expressed in this article are those of Alfonso Siano, Lukman Raimi, Maria Palazzo and Mirela Clementina Panait; "Section 2.3" and "Section 3" are attributed to Lukman Raimi; "Section 2"; "Section 2.2" and "Section 4" are attributed to Mirela Clementina Panait; "Section 2.1" and "Section 5" are attributed to Maria Palazzo; while "Section 1" and "Section 6" is attributed to Alfonso Siano.

**Conflicts of Interest:** The authors declare no conflict of interest.

## Appendix A

**Table A1.** Focus of Selected Paper.

| Article | Title | Author | Year | Number of Factors Affecting Mobile Banking | Type of Specific Factors Affecting Mobile Banking |
|---|---|---|---|---|---|
| 1 | Toward an Understanding of Behavioural Intention to Use Mobile Banking | Luarn & Lin | 2005 | 3 | Perceived credibility, self-efficacy and financial cost |
| 2 | M-Commerce Implementation in Nigeria: Trends and Issues | Ayo, Ekong, Fatudimu & Adebiyi | 2007 | 4 | Patronage, quality of cell phones, lack of basic infrastructure and security issues |
| 3 | Internet Diffusion in Nigeria: is the 'Giant of Africa' waking up? | Muganda, Bankole & Brown | 2008 | 1 | Infrastructure |
| 4 | Mobile Commerce User Acceptance Study in China | Min & Qu | 2008 | 7 | Culture, user satisfaction, trust, privacy protection, quality, experience, and cost |
| 5 | Mobile phone technology in banking system: Its economic effect | Anyasi & Otubu | 2009 | 3 | Convenience, accessibility and affordability. |
| 6 | An Empirical Investigation of the Level of Users' Acceptance of E-Banking in Nigeria. | Oni, Aderonke & Ayo | 2010 | 6 | Convenience, ease of use, time saving, privacy, appropriateness for their transaction needs, and network security |
| 7 | Mobile phones and economic development in Africa | Aker & Mbiti | 2010 | 3 | Ease of use, fast services and reduced communication costs |
| 8 | Mobile banking adoption in Nigeria | Bankole, Bankole & Brown | 2011 | 1 | Cultural Values |
| 9 | An exploratory study on adoption of electronic banking: underlying consumer behaviour and critical success factors: case of Nigeria | Aliyu, Younus & Tasmin | 2012 | 6 | Accessibility, reluctance to change, cost/price, security concern, ease of use, and awareness |

**Table A1.** *Cont.*

| Article | Title | Author | Year | Number of Factors Affecting Mobile Banking | Type of Specific Factors Affecting Mobile Banking |
|---|---|---|---|---|---|
| 10 | Going cashless: Adoption of mobile banking in Nigeria | Njoku & Odumeru | 2013 | 7 | relative advantage, complexity, compatibility, observability, trialability, age and educational background |
| 11 | Global financial development report 2014: Financial inclusion | World Bank | 2013 | 2 | Economic growth and poverty alleviation |
| 12 | An investigative study on factors influencing the customer satisfaction with e-banking in Nigeria | Balogun, Ajiboye & Dunsin | 2013 | 1 | Quality of the service |
| 13 | Impact of mobile banking on service delivery in the Nigerian commercial banks. | Adewoye | 2013 | 4 | Transactional convenience, savings of time, quick transaction alert and save of service cost |
| 14 | Financial inclusion in Africa | Triki & Faye | 2013 | | Broadening access, greater household savings, capital for investment, expansion of class of entrepreneurs, and human capital investment |
| 15 | The opportunities of digitizing payments | Klapper & Singer | 2014 | 2 | Access and Participation |
| 16 | International remittances and financial inclusion in Sub-Saharan Africa | Aga & Peria | 2014 | 1 | Increases the probability of households opening bank accounts |
| 17 | Mobile phone banking in Nigeria: benefits, problems and prospects | Agwu & Carter | 2014 | 4 | Cost of maintenance, Users' education, poverty and infrastructure availability. |
| 18 | Financial inclusion and innovation in Africa | Beck, Senbet & Simbanegavi | 2015 | 3 | Inclusive growth, financial deepening and access |
| 19 | Financial Inclusion: Can It Meet Multiple Macroeconomic Goals? | Sahay, Cihak, M & N'Diaye | 2015 | 3 | Access to credit, Savings and Economic growth |
| 20 | Can Islamic Banking Increase Financial Inclusion? | Ben Naceur, Barajas & Massara | 2015 | 3 | Access to Islamic banking products, improved savings, investment |
| 21 | User adoption of online banking in Nigeria: A qualitative study | Tarhini, Mgbemena, Trab & Masa' Deh | 2015 | 3 | Security, religion and culture |
| 22 | The determinants of financial inclusion in Africa | Zins & Weill | 2016 | 4 | Gender, economic status, education and age influence FI |
| 23 | Mobile banking–adoption and challenges in Nigeria | Agu, Simon & Onwuka | 2016 | 5 | Handset operability, Security, Scalability and reliability, Geographic distribution and Age |
| 24 | Financial inclusion in Africa: evidence using dynamic panel data analysis. | Gebrehiwot & Makina | 2016 | 3 | GDP per capita, mobile infrastructure and remoteness |
| 25 | Analysis of the determinants of financial inclusion in Central and West Africa | Soumaré, TchanaTchana & Kengne | 2016 | 9 | Gender, education, age, income, residence area, employment status, marital status, household size and degree of trust in financial institutions |
| 26 | Is the rise of Pan-African banking the next big thing in Sub-Saharan Africa | PWC | 2017 | 2 | Withdrawal of several Western banks and intra-regional trade linkages |

**Table A1.** *Cont.*

| Article | Title | Author | Year | Number of Factors Affecting Mobile Banking | Type of Specific Factors Affecting Mobile Banking |
|---|---|---|---|---|---|
| 27 | Mobile banking in Sub-Saharan Africa: setting the way towards financial development. | Rouse &Verhoef | 2017 | 2 | Extension of remote rural locations and introduction of innovative products |
| 28 | What determines financial inclusion in Sub-Saharan Africa? | Chikalipah | 2017 | 1 | Illiteracy is the major hindrance to FI |
| 29 | Financial inclusion, entry barriers, and entrepreneurship: evidence from China | Fan & Zhang | 2017 | 3 | Mitigation of credit constraints, boosting entrepreneurial activities and reducing information asymmetry in financial transactions |
| 30 | Determinants of financial inclusion in Sub-Sahara African countries | Oyelami, Saibu & Adekunle | 2017 | 2 | Demand side factors (level of income and literacy) and Supply side factors (Interest rate and bank innovation proxy by ATM usage). |
| 31 | An assessment of the impact of mobile banking on traditional banking in Nigeria | Khan & Ejike | 2017 | 4 | Good knowledge of mobile devices, access to mobile banking, convenience and satisfaction of usage |
| 32 | The effect of mobile banking on the performance of commercial banks in Nigeria | Bagudu, Mohd Khan & Roslan | 2017 | 1 | More access to mobile handsets |
| 33 | Financial inclusion as a tool for sustainable development | Voica | 2017 | 3 | Sustainable development, Consumer protection and economic literacy |
| 34 | The effect of financial inclusion on welfare in sub-Saharan Africa: Evidence from disaggregated data | Tita & Aziakpono | 2017 | 3 | Increase in formal opening of bank accounts, financial infrastructure and economic activities |
| 35 | Infrastructure deficiencies and adoption of mobile money in Sub-Saharan Africa | Mothobi & Grzybowski | 2017 | 2 | physical infrastructure and level of income |
| 36 | Mobile Money and Financial Inclusion in Sub-Saharan Africa: the Moderating Role of Social Networks | Bongomin, Ntayi, Munene & Malinga | 2018 | 1 | Existence of social networks of strong and weak ties among mobile money users |
| 37 | Can mobile money help firms mitigate the problem of access to finance in Eastern sub-Saharan Africa? | Gosavi | 2018 | 2 | Access to finance, or lines of credit |
| 38 | EFInA Access to Financial Services in Nigeria 2010–2018 survey | EFInA | 2018 | 4 | Number of banked population, awareness & knowledge, institutional exclusion and affordability |
| 39 | The Global Findex Database 2017: Measuring financial inclusion and the FinTech revolution | Demirguc-Kunt, Klapper, Singer, Ansar & Hess | 2018 | 0 | |
| 40 | Financial Inclusion and Per Capita Income In Africa: Bayesian VAR Estimates. | Alenoghena | 2019 | 3 | Per capital incomes, deposit interest rate and the internet |
| 41 | M-PESA and Financial Inclusion in Kenya: Of Paying Comes Saving? | Van Hove & Dubus | 2019 | 2 | Phone owners, Better educated |
| 42 | Migrant remittances and financial inclusion among households in Nigeria. | Ajefu & Ogebe | 2019 | 2 | Receipt of remittances increases the use of formal financial services and migrant networks |
| 43 | The Impact of Mobile Money on the Financial Performance of the SMEs in Douala, Cameroon | Talom & Tengeh | 2019 | 3 | Access to the internet, cost and efficiency |

**Table A1.** *Cont.*

| Article | Title | Author | Year | Number of Factors Affecting Mobile Banking | Type of Specific Factors Affecting Mobile Banking |
|---|---|---|---|---|---|
| 44 | Digitising Financial Services: A Tool for Financial Inclusion in South Africa? | Shipalana | 2019 | 3 | Tackle poverty, promote inclusive development and address the SDGs |
| 45 | See the best Nigerian mobile banking apps in H1 2019 | Benson | 2019 | 2 | Access to mobile device, network connection |
| 46 | Financial Inclusion and Achievements of Sustainable Development Goals (SDGs) | Ma'ruf &Aryani | 2019 | 2 | Achievement of SGDs and poverty alleviation |
| 47 | Financial inclusion and sustainable development in Nigeria. | Soyemi, Olowofela & Yunusa | 2019 | 6 | Accessibility, reluctance to change, cost/price, security concern, ease of use, and awareness |
| 48 | Financial inclusion in sub-Saharan Africa: Recent trends and determinants | Asuming, Osei-Agyei, L.G. & Mohammed | 2019 | 6 | Age, education, gender, wealth, growth rate of GDP and access to financial institutions |
| 49 | Enhancing Financial Inclusion in ASEAN: Identifying the Best Growth Markets for Fintech | Loo | 2019 | 4 | Commercial bank branches, Demand deposit from the rural areas, loan to rural areas and human capital development |
| 50 | Social and Financial Inclusion through Nonbanking Institutions: A Model for Rural Romania. | Yue, Cao, Duarte, Shao & Manta | 2019 | 3 | Access to financial services, communication technologies, digital mobile platforms |
| 51 | Do mobile phones, economic growth, bank competition and stability matter for financial inclusion in Africa? | Chinoda & Kwenda | 2019 | 4 | Mobile phones, economic growth, bank competition and stability impact financial inclusion |
| 52 | Financial Inclusion Condition of African Countries | Chinoda & Kwenda | 2019 | 2 | Access and usage factors affect financial inclusion |
| 53 | Mobile telephony, financial inclusion and inclusive growth | Abor, Amidu & Issahaku | 2019 | 3 | Mobile penetration, pro-poor development and improved livelihoods |
| 54 | Financial Inclusion in Ethiopia: Is It on the Right Track? | Berhanu Lakew & Azadi | 2020 | 3 | Barriers are preference for informal saving club, unemployment and low income |
| 55 | Readiness for banking technologies in developing countries | Berndt, Saunders & Petzer | 2020 | 2 | Access to innovative banking technologies and technology readiness of the people |
| 56 | Financial Inclusion | World Bank | 2020 | 3 | Quality of life, poverty reduction, facilitating investments in health, education, and businesses |
| 57 | Financial exclusion in OECD countries: A scoping review | Caplan, Birkenmaier & Bae | 2020 | 6 | Dominant issues covered in FI are conceptualization, contributors, and impacts of FI. Less covered are measurement, prevention, and contemporary practice trends in financial exclusion. |
| 58 | Financial inclusion-and the SDGs | UN Capital Development Fund | 2020 | 3 | Promotes investment, consumption and resource mobilization |

Source: authors, based on the summary of the reviewed literature.

**Appendix B. Year-Wise Distribution of Research Publications**

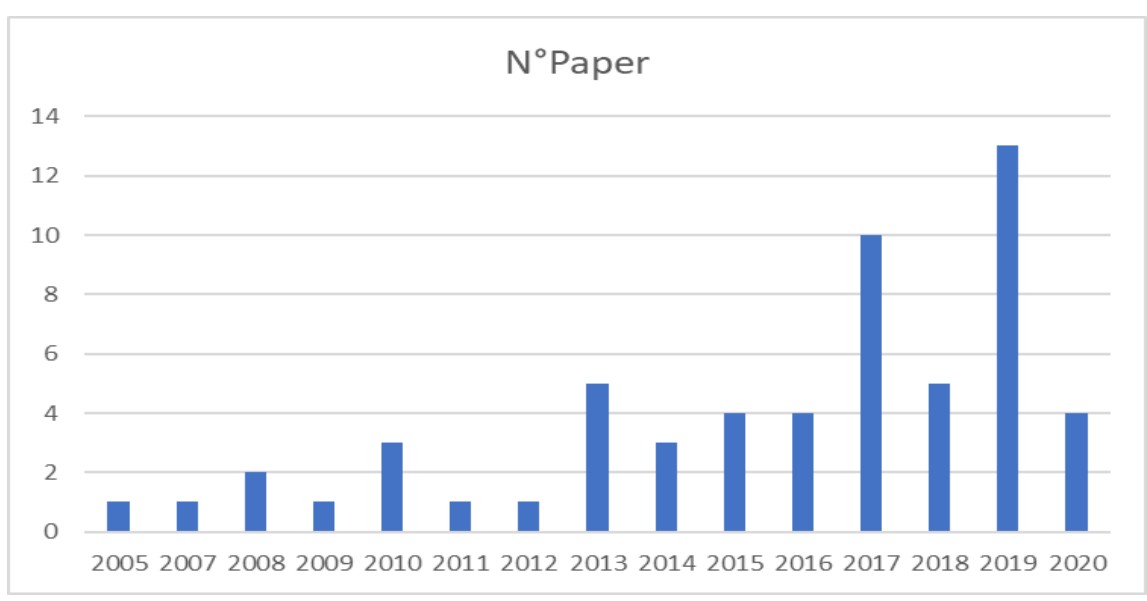

**Figure A1.** Source: authors, based on the summary of the reviewed literature.

**Appendix C. Year-Wise Distribution of Specific Factors Affecting Mobile Banking Explored by Former Publications**

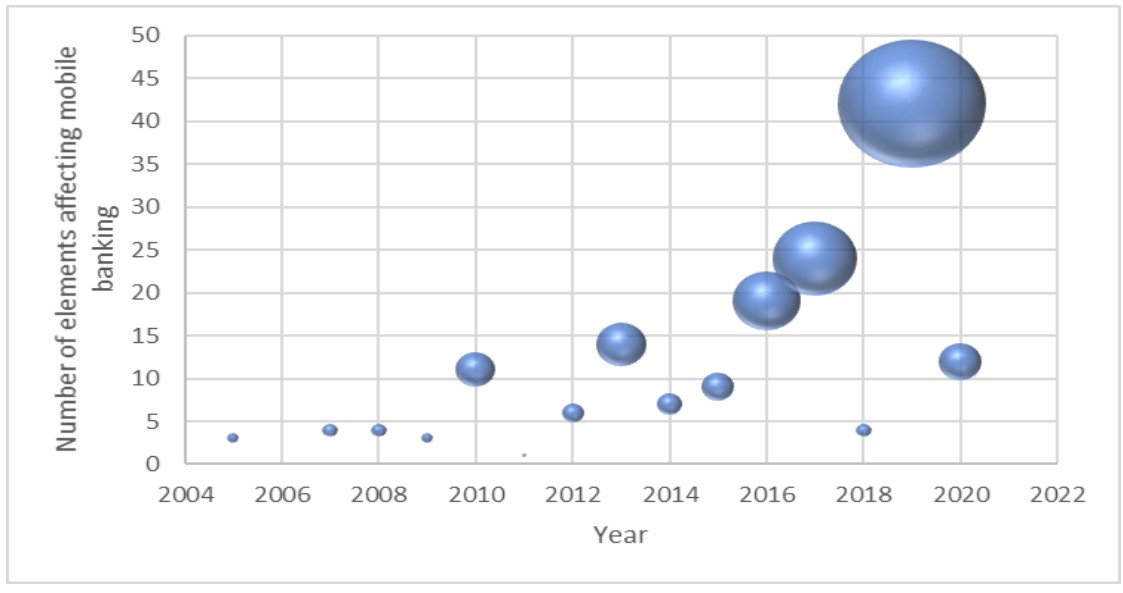

**Figure A2.** Source: authors, based on the summary of the reviewed literature.

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
