# Peer review of "Mobile Banking: An Innovative Solution for Increasing Financial Inclusion in Sub-Saharan African Countries: Evidence from Nigeria"

_sustainability, doi:10.3390/su122310130_

Round 1

Reviewer 1 Report

Estimeed colleagues,

When I saw the title of your articles, I was intrigued because I agree with you the the topic of mobile banking and financial inclusion is indeed interesting. On the positive side, I did find some valuable ideas for banking / finance professionals in your implications section. Similarly, the brief conceptual framework on financial inclusion as well as the description of mobile Banking is quite good and sufficient for such an article.

However, I do have a major concern regarding your paper: I do not see a connection between empirical evidence and your results and conclusions/implications.

Part of the problem is the insufficiently detailed description of the methodology you applied. I would strongly recommend looking at systematic reviews for best practice how this can be transparently described. It is not clear how many studies your original search identified, what selection criteria you exactly applied to narrow down the sample and how the content was systematically reviewed and syntesized. This is crucial information for an articale as this which cannot be ommitted. Or to give a different example, in lines 399 to 415, you talk about VIU and make some interesting points. But what are the empirical results sustaining your claims? What data did you find for instance to argue that “mobile banking is seen by customers as feasible…”?At this point it is simply not transparent enough to follow what you did.

A related point on methodology is that it is not clear in your paper what makes your approach a meta-sysnthesis and not just a summary of the literature? You might be able to show this, but the current version reads to some extent as a summary of existing literature with some additional implications from the authors while it is actually not clear on which evidence you base these recommendations. It might be just a minor mistake that in the conclusion, you call the methodology a meta-analysis, which is definitely the wrong term. Yet, it is another indication that the part of the methodology needs some substantial improvements.

I would recommend starting more systematicall with a Evidence Gap Map (EGM) that show in which fields we actually have evidence. Just as in good practice (you may want to consult with e.g. 3ie on that), you could then use the EGM as a framework to decide where do conduct a systematic review on this topic. I would also suggest that you broaden your search beyond peer reviewed journal articles in order to find more evidence, in a later stage you could then even weight grey literature differently than peer reviewed articles. I would also in line with systematic reviews, suggest broadening your search terms. For instance, terms as in-debtedness, microfinance, digital financial services or even the brand names of mobile banking providers would be crucial. The search terms you mentioned are definitely too few for yielding a sufficiently wide body of literature.

In sum, I think you would have to improve the methodology as well as its decription so that you come up with a more convincing empirical basis for your paper. And then your assertions should be grounded in evidence and those for which you do not have evidence should be taken out. Just to give one example, you talk about the role of opinion leaders to favor adoption. Where do you have this from? It might be correct, but where is the evidence? But then you go on and recommend some forms of compensation by financial institutions for these opinions leaders in form of easy credits. There is no evidence for this and it is furthermore quite questionable from an ethical point of view.

In addition to concerns about the methodology, I am not convinced about part of the conceptual framework relating to mobile banking and financial inclusion. Mobile banking is usually described as the mobile version of financial services for clients of banks. In this sense, mobile banking is by definition not a tool in SSA for financial inclusion. Instead, I would recommend using the concept of mobile financial services or digital financial services (depending on what you want to look at). There is actually quite some literature out on this topic that I would recommend consulting first, because with a conceptual framework on mobile banking there are some limitations what can be learned for improving financial inclusion. Another detail: you would need to define what it means to be an active user of mobile financial services. In SSA, one of the central problems is that many people register but never become really active users. What do you find in terms of improving registration and what in terms of increasing usage? These are two distinct problems that should not be confunded.

Last but not least, I would recommend a more comparartive design. This would broaden the database of articles, but would also help to identify lessons from other countries. Especially regarding the role of the central bank and regulations, a comparison between Nigeria and Kenya would be worthwhile to explore.

Author Response

We would like to thank you the editor and reviewer for providing detailed constructive comments and enabling us to revise the manuscript. We have addressed all the issues and comments carefully and hope the responses appropriately deal with the concerns raised. All main changes have been clearly highlighted, using the “track changes” function in the manuscript. Moreover, 3 new annexes have been added at the end of the text.

We feel the paper has effectively benefited from your input, thus presenting a clearer contribution to knowledge regarding the issues stated in the paper.

We provide our detailed responses to each issue raised.

Best Regards

The authors

Reviewer 2 Report

The paper touches upon a very interesting topic of financial inclusion in Nigeria. However, the paper is very theoretical and has little new input into science. I would rather classify it as a report, rather than a scientific publication. What needs to be highlighted is the extensive topic presentation as well as great knowledge of the authors on the Nigerian system. However, it would be very beneficial if the authors would propose some more detailed research outcome. 

Author Response

(The authors gave the same response as above.)

Reviewer 3 Report

This is an interesting paper.

Authors should correct grammatical errors.

Areas for future research should be added to section 6.

Author Response

(The authors gave the same response as above.)

Round 2

Reviewer 2 Report

The paper has been revised and a lot of scientific sources have been added. In general this has improved the quality of the paper as well as the annexes are a very important input to the whole scientific picture. All in all the paper applies very good descriptive analysis, but lacks some mathematical approach. Maybe it could be a great basis for another paper.